# Expression of Vascular Endothelial Growth Factor-A (VEGF-A) in Adenocarcinoma and Squamous Cell Cervical Cancer and Its Impact on Disease Progression: Single Institution Experience

**DOI:** 10.3390/medicina59071189

**Published:** 2023-06-23

**Authors:** Ivana Piškur, Zlatko Topolovec, Marina Bakula, Irena Zagorac, Iva Milić Vranješ, Domagoj Vidosavljević

**Affiliations:** 1Cathedra for Obstetrics and Gynecology, Faculty of Medicine, University of Osijek, Cara Hadrijana 10e, HR 31000 Osijek, Croatia; ivana.m.spika@gmail.com (I.P.); topolovec.zlatko@kbco.hr (Z.T.); miliciva@yahoo.com (I.M.V.); 2Clinic for Gynecology and Obstetrics, University Hospital Center Osijek, J. Huttlera 4, HR 31000 Osijek, Croatia; bakula.marina@gmail.com (M.B.); irena.frohlich@gmail.com (I.Z.); 3Department of Pathology and Forensic Medicine, University Hospital Centre Osijek, J. Huttlera 4, HR 31000 Osijek, Croatia; 4Faculty of Dental Medicine and Health, University of Osijek, Crkvena ulica 21, HR 31000 Osijek, Croatia; 5Department of Obstetrics and Gynecology, National Memorial Hospital, Bolnička 5, HR 32000 Vukovar, Croatia

**Keywords:** cervical cancer, VEGF, survival, staging

## Abstract

*Background and Objectives*: The aim of this retrospective study was to determine the difference in VEGF-A expression in adenocarcinoma and squamous cell cervical cancer and to show the influence of VEGF-A expression on clinical, pathological, and therapeutic prognostic factors on the outcome of treatment and the survival of patients. *Materials and Methods*: The study included patients with cervical cancer who were treated in the period from 1 January 2010 to 31 December 2021 at the Clinic for Gynecology and Obstetrics, University Hospital Centre, Osijek. The researchers conducted a retrospective analysis of data from patients’ medical history, along with the pathohistological findings and oncologist findings. The study included 66 patients with cervical cancer (divided into two subgroups of 33 with adenocarcinoma or squamous cell cervical cancer). Diagnosis was based on the pathohistological status and FIGO staging. VEGF-A expression was significantly higher in adenocarcinoma. Subjects with a higher expression of VEGF-A had a significantly higher rate of disease progression and a higher possibility for lethal outcome. *Results*: Statistically significant prognostic factors in bivariate analysis in predicting a negative treatment outcome were: older age, greater depth of stromal invasion, FIGO IIB stage, chemotherapy, and positive lymph nodes. In the multivariate analysis, age and positive lymph nodes were shown to be significant predictors for a negative treatment outcome. *Conclusions*: VEGF-A has shown to be statistically more expressed in adenocarcinoma, which correlates with disease progression, but not statistically significant in multivariate regression analysis as an independent prognostic factor for poor survival of the subjects.

## 1. Introduction

Cervical cancer still ranks fourth in incidence (6.5% of all cancer cases) and mortality (7.7% of all cancer deaths) in women worldwide. The largest share of new cases (70%) and deaths (85%) was recorded in countries with a low or medium level of development [1]. Within Europe, Eastern European countries have the highest incidence rate. The disease often occurs at a younger age. The youngest patients in 2018 were between 20 and 24 years old, and 2/3 of newly diagnosed women were younger than 60 [2]. The average age at diagnosis is 45 to 47 years [3]. The incidence of cervical cancer is associated with infection with human papillomaviruses, especially types 16, 18, 31, and 35 [4]. The World Health Organization (WHO) in cooperation with the International Society of Gynecologic Pathologists (ISGP) revised the classification of cervical cancer into three groups [5,6]. It is currently divided into squamous cell cervical cancer, which occurs in 85–90% of all cervical cancers; adenocarcinoma, which occurs in 10–15% of all cervical cancers; and other epithelial tumors [7,8]. The prognostic significance of VEGF-A (Vascular Endothelial Growth Factor-A) in cervical cancer is still a matter of scientific interest [9]. Angiogenesis is considered an important biological process for the growth of primary cancer as well as metastases [10], and according to the literature, VEGF-A is a significant biomarker that enhances tumor angiogenesis [11,12]. An elevated level of VEGF-A is associated with a higher stage of the disease and with metastases in the lymph nodes [12]. So far, it has been shown that an elevated level of VEGF-A is responsible for poor survival in patients with lung, colorectal, as well as ovarian and endometrial cancer [13,14,15]. However, there are only a few studies that have studied the prognostic significance of VEGF-A in cervical cancer and they are all inconclusive [16]. Initial studies have shown that anti-VEGF therapy promotes vascular regression and inhibits tumors and metastases [17]. Previous studies described an elevated level of VEGF-A in cervical cancer, but none of them showed the difference in VEGF-A expression in adenocarcinoma and in squamous cell cervical cancer. It is possible that VEGF-A is more significantly elevated in adenocarcinoma than in squamous cell cervical cancer, and it could be related to its more aggressive behavior and poor survival, as well as poor response to adjuvant therapy [10,15,17].

## 2. Materials and Methods

The study included 66 patients treated for cervical cancer, an equal number of adenocarcinomas and squamous cell cervical cancers, diagnosed and treated in the period from 1 January 2010 to 31 December 2021 at the Clinic for Gynecology and Obstetrics, University Hospital Centre, Osijek. For observing a medium effect (effect size 0.75) in the difference of numerical variables between two independent groups of subjects, with a significance level of 0.05 and a power of 0.8, the minimum required sample size is 29 subjects per group, i.e., a total of 58 subjects. (Calculation was made using program G Power version 3.1.2, Franz Faul, University of Kiel, Germany).

Subjects were diagnosed before 1 January 2017, and then treated and observed for a period of 60 months, which is considered to be a five-year survival rate. The termination of follow-up may be a result of death from underlying or other diseases or termination of research.

Inclusion criteria were FIGO stage IA-IIB and subjects who were primarily treated surgically. Exclusion criteria were FIGO stage III-IVB and subjects who were primarily treated with nonadjuvant oncologic therapy (Table 1).

The aims of this research were to examine the differences in the expression of VEGF-A in adenocarcinoma and squamous cell cervical cancer and to examine the independence of VEGF-A as a prognostic and diagnostic factor in cervical cancer.

The following prognostic factors were analyzed:CLINICAL FACTORS
Age of subjects (at time of diagnosis)FIGO stage (following the revision of FIGO classification from 2018)
PATHOLOGICAL FACTORSHistological type of tumorHistological gradeDepth of stromal invasion (shown in millimeters and divided into groups: <5 mm, 1–5 mm, >5 mm)Invasion of lymphovascular spacesStatus of lymph nodesExpression of VEGF-A in tumor cells
POSTOPERATIVE THERAPY FACTORS
Adjuvant radiotherapy(a)pelvic irradiation(b)brachytherapy(c)combination of pelvic irradiation and brachytherapyChemotherapy

The outcome of this type of treatment was expressed as complete remission (with no signs of active disease), partial remission (with a positive PAP test and no other signs of residual disease), or progression (with a positive PAP test and signs of metastatic disease). The evaluation of the oncologic treatment was determined using a PAP test and a CT scan of the chest, abdomen, and pelvis, or possibly a PET CT scan.

### 2.1. Immunochistochemistry

Selected paraffin blocks from each case were cut at 4 μm thickness, deparaffinized in xylen at a temperature of 72 °C, and then stained using the standard hemalaun-eosin method. For the objectivity of the findings, the histological slides were examined by two independent pathologists.

Immunohistochemical staining for VEGF-A expression in tumor cells was performed using a VENTANA BenchMark Ultra device. When entering the program for the VEGF antibody catalog number sc-7269 into the Ventana BenchMark Ultra system, their protocol was entered into the device. The cut and dried slide with a paraffin section was placed in the device, and fully automated deparaffinization was performed in it using EZ Prep concentrate (LOT 206236-01, REF 950-102), which was diluted 1:10, placed in the container, and the device applied it to the slide with a pump. VEGF-A was performed using the Ultra View Universal DAB Detection kit (REF 760-500, LOT G07369). Following deparaffinization, rehydration, and blockage of endogenous peroxidase, the slides were treated with antigen retrieval solution followed by incubation with VEGF antibodies (VEGF-A antibody (C1): sc-7269 is a mouse monoclonal antibody manufactured by Santa Cruz biotechnology. The LOT of the antibody is C0922). Antibody binding was visualized using the iView DAB Detection kit (760-091, Ventana), an indirect biotin streptavidin system for detecting mouse primary antibodies. Placental tissue was used as a positive test. For the purpose of this study, we considered it positive if 10% of tumor cells showed cytoplasmic staining of any intensity. Samples were analyzed under 4×, 10×, 20×, and 40× magnification objectives. VEGF-A 3+ staining was seen as intense cytoplasmic staining, immediately visible at 4× magnification, and confirmed at 10× magnification. Moderate cytoplasmic 2+ staining was weaker and thinner than 3+ staining, visible at 10× magnification and confirmed at 20× magnification, and weak cytoplasmic 1+ staining required 20× and/or 40× magnification and manifested as extremely weak, thin cytoplasmic dyeing. For the purposes of this study, the positive VEGF-A values were divided into groups as follows: low (+), medium (++), and high (+++) (Figure 1).

### 2.2. Statistical Methods

Categorical data were represented by absolute and relative frequencies. Numerical data were described by the arithmetic mean and standard deviation in the case of distributions that follow the normal, and in other cases by the median and the limits of the interquartile range.

The association of categorical variables was tested with the X2 test and, if necessary, with Fisher’s exact test. The normality of the distribution of numerical variables was tested with the Shapiro–Wilk test. Differences normally distributed by numerical variables between two independent groups were tested with the Student’s *t* test, and in case of deviation from the normal distribution, the Mann–Whitney U test (with the stated difference and 95% confidence interval of the difference). Survival analysis was completed with the Cox regression test on incomplete data. Variables that showed a significant association with the criterion were included in the multivariate model. Logistic regression (bivariate and multivariate) was used to examine the significance of independent factors in predicting the occurrence of cervical cancer of glandular origin. The association of normally distributed numerical variables was assessed using Pearson’s correlation coefficient r, and in case of deviation from normal distribution, with Spearman’s correlation coefficient ρ (rho). All ***p*** values were two-sided. The significance level was set at α = 0.05. Statistical analysis was completed using the statistical program MedCalc (version 16.2.0, MedCalc Software bvba, Ostend, Belgium).

## 3. Results

The research involved 66 subjects of whom 33 (50%) each had adenocarcinoma or squamous cell cervical cancer. The expression of the VEGF-A +++ included 27 (41 %) patients, significantly more of them, 23 (70 %), with adenocarcinoma (Fisher’s exact test, *p* < 0.001). Using FIGO staging, no significant differences were observed in the distribution of patients with regard to the type of cancer (Fisher’s exact test, *p* = 0.007). Patients with adenocarcinoma had a histological grade III, which was significantly higher compared to patients with squamous cell cervical cancer (Fisher’s exact test, *p* = 0.03). Lymphovascular invasion was observed in 44 (67%) patients, 24 (36%) patients had pelvic radiotherapy, 2 (3%) patients had brachitherapy, and 14 (21%) patients had chemotherapy. Complete remission occurred in 52 (79%) patients, significantly more often in patients with squamous cell cervical cancer, while disease progression was more significant in patients with adenocarcinoma (Fisher’s exact test, *p* = 0.008). In total, 59 (89%) patients survived, showing a borderline significance in relation to the type of cancer (Table 2).

Patients with disease progression showed significantly higher levels of VEGF-A +++ (Fisher’s exact test, *p* = 0.02), were identified as FIGO stage II B (Fisher’s exact test, *p* = 0.03), had positive lymph nodes (Fisher’s exact test, *p* < 0.001), had pelvic radiotherapy (Fisher’s exact test, *p* < 0.001), and had a significantly higher death outcome (Fisher’s exact test, *p* < 0.001) (Table 3).

There is a significant difference in the distribution of treatment outcome in relation to VEGF-A expression. In the case of disease progression, there are significantly more patients with VEGF-A+++ expression (Fisher’s exact test, *p* = 0.02) (Table 4).

The shortest survival is significant in patients with disease progression—median 15 months (95% CI 12–18)—compared to patients with complete or partial regression of the disease (Log rank test, *p* = 0.03) (Table 5).

Bivariate regression analysis observed that survival is influenced by the age of the subjects (OR = 1.14) and positive lymph nodes (OR = 5.41) (Table 6).

Cox multivariate regression analysis (stepwise method) showed that age and positive lymph nodes were the significant models that affect survival (Table 7).

The subjects with a negative treatment outcome were significantly older. The median age was 67 years, ranging from 57 to a maximum of 80 years, compared to the patients who survived (age range from 27 to a maximum of 74 years) (Table 8).

Bivariate logistic regression was used to assess the influence of factors on the probability of partial regression or progression. Significant predictors in bivariate logistic regression in confirming the probability of partial regression or progression are older age (OR = 1.07), greater depth of stromal invasion (OR = 1.2), FIGO II B stage (OR = 9.44), chemotherapy (OR = 4.13), and positive lymph nodes (OR = 10.2) (Table 9).

As a model of multivariate regression analysis, we observed predictors that were significant in bivariate analysis. The model consists of two predictors, age (OR = 1.07) and positive lymph nodes (OR = 9.1). It explains between 25% (according to Cox and Snell) and 38% (according to Negelkerke) of the variance present in partial regression or progression, and correctly classifies 84% of cases (Table 10).

Bivariate logistic regression was used to assess the influence of factors on the probability of a negative outcome. Significant predictors in bivariate logistic regression are older age (OR = 1.18) and positive lymph nodes (OR = 6.5) As a model of multivariate regression analysis, we observed predictors of negative outcome that are significant in bivariate analysis. The model consists of only one predictor, the age of the test subjects, which means older patients have a 1.18 times higher chance of lethal outcome. It explains between 22% (according to Cox and Snell) and 46% (according to Negelkerke) of the variance of negative outcome, and correctly classifies 91% of cases (Table 11 and Table 12).

## 4. Discussion

The study included patients treated for cervical cancer in the period from 2010 to 2021 at the Clinic for Gynecology and Obstetrics, University Hospital Centre, Osijek. The study included 66 subjects, i.e., an equal number of adenocarcinomas and squamous cell cervical cancers. 

The inclusion criteria were FIGO stage IA-IIB and subjects who were primarily treated surgically. The aim of this research was to show the difference in VEGF-A expression between the two examined groups using immunohistochemical methods and to demonstrate the independence of VEGF-A as a factor in the prognosis and diagnosis of subjects suffering from these two types of cancer. This research showed that 27 subjects with squamous cell cervical cancer had a higher expression of VEGF-A (+++), and significantly more of them, 23 (70%), with adenocarcinoma. Another study showed higher expression of VEGF-A in squamous cell cervical cancer in comparison to adenocarcinoma [18].

Using FIGO staging, no significant differences were observed in the distribution of subjects with regard to the type of cancer, which correlates with the above-mentioned research [18].

This study showed that 6 (18%) subjects with adenocarcinoma were histological grade III, meaning poorly differentiated cancer, representing significantly more than those with squamous cell cervical cancer. In the aforementioned research, the histological grade was not statistically significant [18].

Lymphovascular invasion occurred in 23 subjects with squamous cell cervical cancer and in 21 subjects with adenocarcinoma, which was not statistically significant. In contrast to our research, Lee et al. showed that there is a statistically significant difference in the invasion of lymphovascular spaces in adenocarcinoma in contrast to squamous cell cervical cancer [19].

Twelve subjects with adenocarcinoma had statistically significantly more positive lymph nodes in contrast to two subjects with squamous cell cervical cancer.

Complete remission occurred in 52 (79%) subjects, significantly more often in subjects with squamous cell cervical cancer, while disease progression was more significant in subjects with adenocarcinoma, which correlates with Lee et al. [19].

This research showed that subjects with disease progression had a significantly higher expression of VEGF-A (+++), positive lymph nodes, received pelvic radiotherapy more often, and were significantly more likely to have a fatal outcome.

There is a significant difference in the distribution of treatment outcomes in relation to VEGF-A expression. In the case of disease progression, there are significantly more patients with VEGF-A+++. In this study, in a bivariate analysis, positive lymph nodes and age in subjects with adenocarcinoma proved to be a significant combination that affects the survival of the subjects. For comparison, in the research of Zhou et al., the number of positive lymph nodes in adenocarcinoma did not show its prognostic value [20].

Our research showed that complete remission occurred in 52 (79%) subjects, significantly more often in subjects with squamous cell cervical cancer, while disease progression was more significant in patients with adenocarcinoma. 

Islam et al. in their research showed a three-year trend without progression in subjects with squamous cell cervical cancer, as opposed to adenocarcinoma, although they did not find it statistically significant [21].

Subjects with disease progression, which were shown in our work to have an increased expression of VEGF-A, were significantly older. Previously mentioned research investigated the expression of VEGF-A in younger and older patients and did not show a statistically significant difference [21].

As a model of multivariate regression analysis, we observed predictors that were significant in bivariate analysis. The model consists of two predictors, age and positive lymph nodes, which also proved to be independent factors in predicting lethal outcome. After searching the available literature, no relevant study was found that examined the influence of VEGF-A on prognostic factors between these two types of cancer.

## 5. Conclusions

VEGF-A expression is statistically significantly higher in adenocarcinoma. Despite a small number of subjects, patients with a higher expression of VEGF-A have a statistically significantly higher rate of disease progression and a higher chance of death outcome. Statistically significant prognostic factors in bivariate analysis in predicting a negative treatment outcome were older age, greater depth of stromal invasion, FIGO IIB stage, history of chemotherapy, and positive lymph nodes. In the multivariate analysis, age and positive lymph nodes were shown to be significant predictors for a negative treatment outcome. Further extended research is needed.

## Figures and Tables

**Figure 1 medicina-59-01189-f001:**
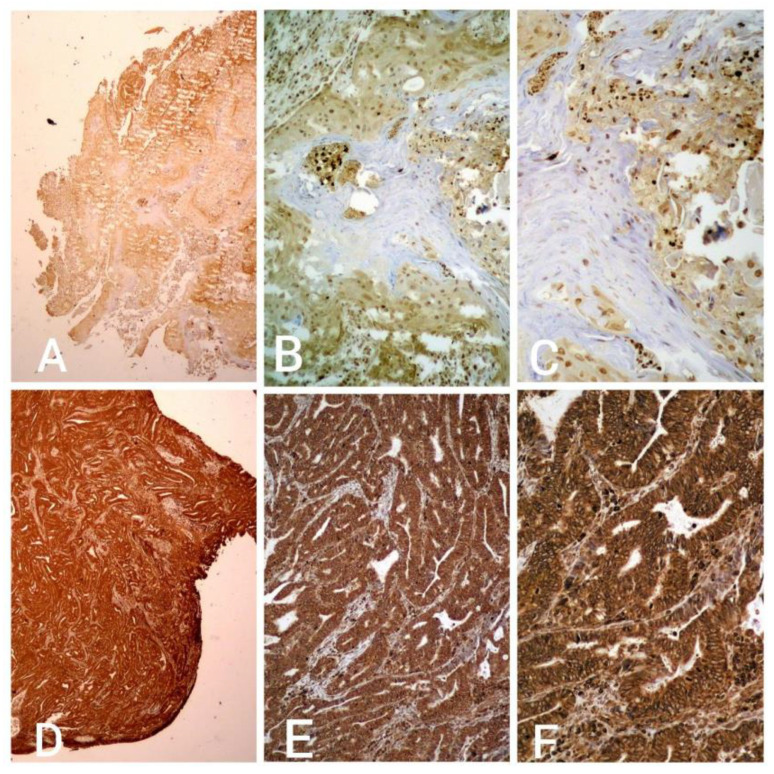
VEGF-A expression in squamous cell cervical cancer (**A**) 1+, 2× magnification; (**B**) 1+, 10× magnification; and (**C**) 1+, 20× magnification. Expression of VEGF-A in adenocarcinoma (**D**) 3 +++, 2× magnification; (**E**) 3 +++, 10× magnification; and (**F**) 3 +++, 20× magnification (pictures from personal archive).

**Table 1 medicina-59-01189-t001:** International Federation of Gynecology and Obstetrics (FIGO) staging of cancer of the cervix uteri (2018).

Stage	Description
I	The carcinoma is strictly confined to the cervix (extension to the uterine corpus should be disregarded)
IA	Invasive carcinoma that can be diagnosed only by microscopy, with maximum depth of invasion <5 mm
IA1	Measured stromal invasion <3 mm in depth
IA2	Measured stromal invasion ≥3 mm and <5 mm in depth
IB	Invasive carcinoma with measured deepest invasion ≥5 mm (greater than Stage IA), lesion limited to the cervix uteri
IB1	Invasive carcinoma ≥5 mm depth of stromal invasion, and <2 cm in greatest dimension
IB2	Invasive carcinoma ≥2 cm and <4 cm in greatest dimension
IB3	Invasive carcinoma ≥4 cm in greatest dimension
II	The carcinoma invades beyond the uterus, but has not extended onto the lower third of the vagina or to the pelvic wall
IIA	Involvement limited to the upper two-thirds of the vagina without parametrial involvement
IIA1	Invasive carcinoma <4 cm in greatest dimension
IIA2	Invasive carcinoma ≥4 cm in greatest dimension
IIB	With parametrial involvement but not up to the pelvic wall
III	The carcinoma involves the lower third of the vagina and/or extends to the pelvic wall and/or causes hydronephrosis or nonfunctioning kidney and/or involves pelvic and/or para-aortic lymph nodes
IIIA	The carcinoma involves the lower third of the vagina, with no extension to the pelvic wall
IIIB	Extension to the pelvic wall and/or hydronephrosis or nonfunctioning kidney (unless known to be due to another cause)
IIIC	Involvement of pelvic and/or para-aortic lymph nodes, irrespective of tumor size and extent (with r and p notations)
IIIC1	Pelvic lymph node metastasis only
IIIC2	Para-aortic lymph node metastasis
IV	The carcinoma has extended beyond the true pelvis or has involved (biopsy-proven) the mucosa of the bladder or rectum. (A bullous edema, as such, does not permit a case to be allotted to Stage IV).
IVA	Spread to adjacent pelvic organs
IVB	Spread to distant organs

**Table 2 medicina-59-01189-t002:** Baseline characteristics.

	Number (%) of Subjects			*p **
	Squamous Cell Cervical Cancer	Adenocarcinoma	Total	
Expression of VEGF-A				
+	5 (15)	2 (6)	7 (11)	**<0.001**
++	24 (73)	8 (24)	32 (48)	
+++	4 (12)	23 (70)	27 (41)	
FIGO stage				
I A	8 (24)	12 (36)	20 (30)	0.28
I B	21 (64)	13 (39)	34 (52)	
II A	1 (3)	3 (9)	4 (6)	
II B	3 (9)	5 (15)	8 (12)	
Histological grade				
G I	7 (21)	12 (36)	19 (29)	**0.03**
G II	25 (76)	15 (45)	40 (61)	
G III	1 (3)	6 (18)	7 (11)	
Lymphovascular space invasion	23 (70)	21 (64)	44 (67)	0.79
Positive lymph node	2 (6)	12 (36)	14 (21)	**0.005**
Radiotherapy				
Pelvic	11 (33)	13 (39)	24 (36)	0.80
Brachytherapy	1 (3)	1 (3)	2 (3)	>0.99
Chemotherapy	8 (24)	6 (18)	14 (21)	0.76
Treatment outcome				
Regression	30 (91)	22 (67)	52 (79)	**0.008**
Partial regression	3 (9)	3 (9)	6 (9)	
Progression	0	8 (24)	8 (12)	
Survival				
Alive	32 (97)	27 (82)	59 (89)	0.05
Dead	1 (3)	6 (18)	7 (11)	

* Fisher’s exact test.

**Table 3 medicina-59-01189-t003:** Baseline characteristics.

	Number (%) of Subjects				*p* *
	Regression	Partial Regression	Progression	Total	
Expression of VEGF-A					
+	6 (12)	0	1 (13)	7 (11)	**0.02**
++	28 (54)	4 (67)	0	32 (48)	
+++	18 (35)	2 (33)	7 (88)	27 (41)	
FIGO stage					
I A	17 (33)	1 (17)	2 (25)	20 (30)	**0.03**
I B	29 (56)	1 (17)	4 (50)	34 (52)	
II A	3 (6)	1 (17)	0	4 (6)	
II B	3 (6)	3 (50)	2 (25)	8 (12)	
Histological grade					
G I	16 (31)	1 (17)	2 (25)	19 (29)	0.58
G II	31 (60)	5 (83)	4 (50)	40 (61)	
G III	5 (10)	0	2 (25)	7 (11)	
Lymphovascular space invasion	32 (62)	5 (83)	7 (88)	44 (67)	0.29
Positive lymph nodes	6 (12)	2 (33)	6 (75)	14 (21)	**<0.001**
Radiotherapy					
Pelvic	12 (23)	5 (83)	7 (88)	24 (36)	**<0.001**
Brachytherapy	1 (2)	1 (17)	0	2 (3)	0.19
Chemotherapy	8 (15)	3 (50)	3 (38)	14 (21)	0.05
Survival					
Alive	52 (100)	5 (83)	2 (25)	59 (89)	**<0.001**
Dead	0	1 (17)	6 (75)	7 (11)	

* Fisher’s exact test.

**Table 4 medicina-59-01189-t004:** Distribution of subjects according to treatment outcome in relation to VEGF-A expression.

	Number (%) of Subjects Expression of VEGF-A				*p* *
	+	++	+++	Total	
Treatment outcome					
Regression	6 (86)	28 (88)	18 (67)	52 (79)	**0.02**
Partial regression	0	4 (13)	2 (7)	6 (9)	
Progression	1 (14)	0	7 (26)	8 (12)	
Total	7 (100)	32 (100)	27 (100)	66 (100)	

* Fisher’s exact test.

**Table 5 medicina-59-01189-t005:** Survival in relation to outcome.

	Number (%) of Deaths	Number (%) of Survivals	Total	Survival		Logrank Test (*p*)
				Mean (Months)	95% CI	
Total survival	7	59	66	55.2	51.9–58.6	
Treatment outcome						
Regression	0	52	52	60	60–60	**<0.001**
Partial regression	1	5	6	55	47–64	
Progression	6	2	8	15	12–18	

CI—confidence interval.

**Table 6 medicina-59-01189-t006:** Survival analysis by Cox regression (bivariate regression).

	ß	*p*	OR	95% CI
Age	0.13	**0.002**	1.14	1.05–1.24
Depth of stromal invasion	0.11	0.09	1.12	0.98–1.27
FIGO stage (IA *)				
I B	0.79	0.48	2.20	0.25–19.7
II A	−10.8	0.96	0	-
II B	1.63	0.18	5.09	0.46–56.2
Cancer (squamous cell *)				
Adenocarcinoma	1.97	0.07	7.20	0.86–60.42
Histological grade (G I *)				
G II	−0.08	0.93	0.93	0.17–5.06
G III	0.29	0.82	1.33	0.12–14.7
Expression of VEGF-A (+ *)				
+	−1.53	0.28	0.21	0.01–3.44
++	0.35	0.75	1.42	0.17–12.2
Chemotherapy	1.01	0.19	2.73	0.61–12.2
Positive lymph nodes	1.69	**0.03**	5.41	1.21–24.3
Lymphovascular space invasion	1.12	0.30	3.06	0.37–25.4
Treatment outcome (regression *)				
Partial regression	15.1	0.97	-	-
Progression	29.9	0.96	-	-

ß—regression coefficient; CI—confidence interval; * reference value.

**Table 7 medicina-59-01189-t007:** Survival analysis by Cox regression (multivariate regression).

	ß	*p*	OR	95% CI
Age	0.13	**0.002**	1.14	1.05–1.24
Positive lymph nodes	1.48	**0.008**	4.39	1.47–13.16

ß—regression coefficient; CI—confidence interval.

**Table 8 medicina-59-01189-t008:** Differences in the age of the patients in relation to the outcome.

	Median (IQR)		*Differences* ^*†*^	95% CI	*p* *
	Alive	Deceased			
Age [years]	44 (37–59)	67 (60–75)	22	12–30	**0.001**

* Mann–Whitney U test; *^†^* Hodges–Lehmann median differences; CI—confidence interval.

**Table 9 medicina-59-01189-t009:** Predicting the probability of partial regression or progression—bivariate regression analysis.

	ß	*p*	OR	95% CI
Age	0.08	**0.005**	1.07	1.02–1.14
Depth of stromal invasion	0.18	**0.003**	1.20	1.06–1.35
FIGO stage (IA *)				
I B	−0.02	0.98	0.98	0.21–4.61
II A	0.64	0.63	1.89	0.4–24.8
II B	2.24	**0.02**	9.44	1.43–62.2
Cancer (squamous cell *)				
Adenocarcinoma	1.61	**0.02**	5.0	1.25–20.1
Histological grade (G I *)				
G II	0.44	0.55	1.55	0.37–6.53
G III	0.76	0.47	2.13	0.27–16.6
Expression of VEGF-A (+ *)				
+	−0.15	0.89	0.86	0.08–9.09
++	1.09	0.34	3.0	0.31–28.8
Chemotherapy	1.41	**0.03**	4.13	1.13–15.1
Positive lymph nodes	2.32	**<0.001**	10.2	2.63–39.7
Lymphovascular space invasion	1.32	0.10	3.75	0.76–18.5

ß—regression coefficient; CI—confidence interval; * reference value.

**Table 10 medicina-59-01189-t010:** Prediction of the probability of poor treatment outcome (partial regression or progression) (multivariate regression analysis).

	ß	*p*	OR	95% CI
Age	0.07	**0.02**	1.07	1.01–1.14
Positive lymph nodes	2.21	**0.004**	9.13	2.04–40.9
Constant	−5.65	**<0.001**		

ß—regression coefficient; CI—confidence interval.

**Table 11 medicina-59-01189-t011:** Predicting the probability of a negative outcome (death)—bivariate regression analysis.

	ß	*p*	OR	95% CI
Age	0.17	**0.005**	1.18	1.05–1.33
Depth of stromal invasion	0.12	0.09	1.12	0.98–1.29
FIGO stage (IA *)				
I B	0.93	0.42	2.53	0.26–24.4
II A	−16.8	0.99	0	-
II B	1.85	0.16	6.33	0.48–82.7
Cancer (squamous cell *)				
Adenocarcinoma	1.96	0.08	7.11	0.81–62.8
Histological grade (G I *)				
G II	−0.06	0.95	0.94	0.16–5.47
G III	0.35	0.79	1.42	0.11–18.6
Expression of VEGF-A (+ *)				
+	−1.64	0.27	0.19	0.01–3.54
++	0.31	0.79	1.36	0.13–14.01
Chemotherapy	1.19	0.16	3.27	0.64–16.8
Positive lymph nodes	1.87	**0.03**	6.5	1.26–33.8
Lymphovascular space invasion	1.20	0.28	3.32	0.37–29.4

ß—regression coefficient; CI—confidence interval; * reference value.

**Table 12 medicina-59-01189-t012:** Prediction of the probability of a negative outcome (death) (multivariate regression analysis of the stepwise method).

	ß	*p*	OR	95% CI
Age	0.17	**0.005**	1.18	1.05–1.33
Constant	−11.9	0.002		

ß—regression coefficient; CI—confidence interval.

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
