# Peer review of "Expression of Vascular Endothelial Growth Factor-A (VEGF-A) in Adenocarcinoma and Squamous Cell Cervical Cancer and Its Impact on Disease Progression: Single Institution Experience"

_medicina, 2023, doi:10.3390/medicina59071189_

Round 1
Reviewer 1 Report
The authors present a clear introduction to the aims of the study.
The sample size, although explained clearly in the statistical portion, is low as it does not clearly support the results of the study as claimed by the authors.
An erroneous statement in the manuscript states "Out of a total of 27 subjects with expression of the VEGF-A +++, significantly more of them (88%) had disease progression". This statement is wrong as only 7/27 patients (approx 26%) with progression had VEGF-A. The rest of them, with high VEGF expression actually did not show progression, while 1/8 had low VEGF-A expression. This needs to be corrected wherever the claim is made.
Table 3 is redundant as the same data is presented in Table2. The results section also shows considerably redundant data contained in the Tables included within the manuscript.
The authors describe several clinicopathological features of recruited patients treated for cervical cancer. However, there is no mention of HPV in the study since the presence of specific HPV subtypes is known to be associated with specific cellular subtypes.
Overall, the study results lack strength and fail to clearly establish VEGF A as an independent prognostic indicator.
The manuscript requires thorough revision in terms of language, as there are many spelling and grammatical errors which make it difficult for readers to comprehend the meaning of sentences.
Author Response
Thank You very much for Your effort. Your comments have been accepted and additional explanations have been given.

Reviewer 2 Report
The authors have produced a very interesting pathological and epidemiology study on Cervical cancer, showing that more VEGF-A expression in adenocarcinoma correlating with disease progression, but not statistically significant in multivariate regression analysis as an independent prognostic factor for poor survival. Although with small group of 66 patients, this study still contains clinical merit.
1. Raw data of VEGF-A IHC should be provided, at least, the representative pictures of VEGF-A with different scores should be in the article.
2. Raw data could be searched by the readers, which is the merit of this manuscript.
Author Response

(The authors gave the same response as above.)

Reviewer 3 Report
The paper "EXPRESSION OF VASCULAR ENDOTHELIAL GROWTH FACTOR A (VEGF-A) IN ADENOCARCINOMA AND SQUAMOUS CELL CERVICAL CARCINOMA AND ITS IMPACT ON DISEASE PROGRESSION: SINGLE INSTITUTION EXPERIENCE" by authors Dr.Ivana Piskur and their coworkers is fascinating. Over all the study is much needed and made an interesting analysis along with the pathohistological findings and oncologist findings was made and the study included 66 patients with cervical cancer. The scientists conclude that despite having a smaller number of patients, those people who express more VEGF-A have a statistically significant greater rate of illness progression and a higher likelihood of dying as a result.
I recommend the manuscript for publication after minor corrections
Minor comments:
There are some typo through out the manuscript, the authors should rectify it accordingly.
If possible draw picture illustrating the stages of FIGO
Minor corrections need to be done
Author Response

(The authors gave the same response as above.)

Reviewer 4 Report
Overall, I find this manuscript to be a valuable contribution to the field of cervical cancer research. The study findings provide important insights into the role of VEGF-A expression in cervical cancer, and the study methodology is appropriate for the research question. I recommend that the manuscript be accepted for publication, with minor revisions to improve the clarity of the language and presentation of the results. However, I first want to know the reply of authors about following comments:
Despite the study's thorough retrospective analysis of 66 patients with cervical cancer, the sample size is relatively small, and the study's findings may not be generalizable to larger populations. The study's inclusion and exclusion criteria limit the generalizability of the findings to patients with FIGO stage IA-IIB who were primarily treated surgically. The study's retrospective nature may also introduce potential bias in data collection and analysis. Additionally, while the study provides valuable insights into the prognostic significance of VEGF-A expression, it does not explore other potential prognostic factors that may influence the outcome of treatment and survival in patients with cervical cancer.
I find the results of this study to be informative and well-presented. The study provides important insights into the factors that influence the outcome of treatment and survival in patients with cervical cancer.
Overall, the study provides valuable insights into the factors that influence the outcome of treatment and survival in patients with cervical cancer. The results of the study could be useful for clinicians in developing more effective treatment strategies and for researchers in designing future studies focused on improving outcomes for patients with cervical cancer.
I recommend the manuscript be edited by a native person
Author Response

(The authors gave the same response as above.)

Round 2
Reviewer 1 Report
Authors have answered queries raised, appropriately.
The authors have taken necessary steps to improve significantly, the quality of English language.